# Glymphatic System Pathology and Neuroinflammation as Two Risk Factors of Neurodegeneration

**DOI:** 10.3390/cells13030286

**Published:** 2024-02-05

**Authors:** Stanisław Szlufik, Kamila Kopeć, Stanisław Szleszkowski, Dariusz Koziorowski

**Affiliations:** Department of Neurology, Faculty of Health Science, Medical University of Warsaw, 02-091 Warszawa, Poland; kopec.kaamila@gmail.com (K.K.);

**Keywords:** neurodegeneration, glymphatic system, neuroinflammation, BBB opening, Parkinson’s disease, Alzheimer’s disease, AQP4, gut–brain axis

## Abstract

The key to the effective treatment of neurodegenerative disorders is a thorough understanding of their pathomechanism. Neurodegeneration and neuroinflammation are mutually propelling brain processes. An impairment of glymphatic system function in neurodegeneration contributes to the progression of pathological processes. The question arises as to how neuroinflammation and the glymphatic system are related. This review highlights the direct and indirect influence of these two seemingly independent processes. Protein aggregates, a characteristic feature of neurodegeneration, are correlated with glymphatic clearance and neuroinflammation. Glial cells cannot be overlooked when considering the neuroinflammatory processes. Astrocytes are essential for the effective functioning of the glymphatic system and play a crucial role in the inflammatory responses in the central nervous system. It is imperative to acknowledge the significance of AQP4, a protein that exhibits a high degree of polarization in astrocytes and is crucial for the functioning of the glymphatic system. AQP4 influences inflammatory processes that have not yet been clearly delineated. Another interesting issue is the gut–brain axis and microbiome, which potentially impact the discussed processes. A discussion of the correlation between the functioning of the glymphatic system and neuroinflammation may contribute to exploring the pathomechanism of neurodegeneration.

## 1. Introduction to Neurodegeneration

Neurodegeneration is an umbrella term for a group of diseases with highly heterogeneous clinical presentations [1]. Their common denominator is the gradual and progressive loss of neurons, leading to the impairment of the functioning of the nervous system, and, consequently, the occurrence of disease symptoms [2]. The etiology of neurodegenerative diseases is multifactorial. Interactions between environmental and genetic risk factors are expected to play a unique role [3,4]. However, the pathogenesis of neurodegeneration is not yet completely understood. It is widely acknowledged that the significant factor in neurodegenerative disorders is the accumulation of misfolded proteins, which are implicated in the pathogenesis of neurodegeneration [1,4]. Neuroinflammation and the impaired functioning of the glymphatic system, as well as oxidative stress, are involved in this process [2,5,6,7]. Neurodegenerative disorders are one of the greatest challenges in modern neurology. They are a serious socioeconomic problem and, more importantly, a common burden that reduces quality of life and causes mortality, especially among the elderly population [8,9]. However, to lead to the discovery of an intensively sought effective causal therapy, it seems necessary to have a thorough understanding of this cascade of pathological processes occurring in the disease.

## 2. Neuroinflammation in Neurodegeneration

The two most common neurodegenerative diseases are Alzheimer’s disease (AD) and Parkinson’s disease (PD) [1]. Protein aggregation is a characteristic feature of many neurodegenerative disorders, including AD and PD. Misfolded proteins accumulate in the brain as protein aggregates or inclusions and their presence has a toxic effect on cells [10]. These proteins are β-amyloid (Aβ) in AD and α-synuclein (α-Syn) in PD [11]. Importantly, the role of protein aggregates in the process of neuroinflammation in the course of neurodegenerative diseases is emphasized [12,13]. Neuroinflammation, including microgliosis and astrogliosis, is closely related to neurodegenerative diseases, significantly contributing to their pathogenesis and progression [14]. In principle, neuroinflammation is a beneficial process; it initially allows the body to defend itself by controlling and removing pathological stimuli. It also participates in tissue repair with the aim of limiting damage and restoring homeostasis. It is a self-limiting process crucial for its beneficial function. The persistent inflammatory process in neurodegeneration is self-perpetuating, leads to progressive cell damage, and contributes to disease pathology [15]. The central nervous system (CNS) comprises two major cell groups: neurons and glial cells. Glial cells are subdivided into different types, including astrocytes and microglia [16]. Specific phenotypes of astrocytes and microglia are involved in the inflammatory response and play neurotoxic or neuroprotective roles [17]. The phenotypes of microglial cells depend on their activation status. Microglia may acquire neurotoxic functions depending on the type and stage of neurodegenerative diseases. There are also differences in the proportion of phenotypes depending on the progression of neurodegenerative diseases [18]. Both astrocytes and microglia produce inflammatory mediators such as tumor necrosis factor (TNF)-α, interleukin (IL)-1β, and IL-6. Importantly, pro-inflammatory mediators secreted by microglia promote a secondary inflammatory response by activating pro-inflammatory astrocytes [18]. Along with inflammation, neuronal death occurs, intensifying the release of pro-inflammatory mediators and inflammatory processes. Thus, neuroinflammation and neurodegeneration are locked into a vicious circle [19].

## 3. Glymphatic System

A high metabolic rate and the intensive production of metabolic waste accompany the energy demand of the organ. The lymphatic system plays a prominent role in maintaining tissue homeostasis, which is necessary for its proper functioning. One exception is the brain [20,21]. The brain parenchyma is devoid of lymphatic vessels; however, its energy demand is one of the highest, amounting to approximately 20% of the body’s energy consumption [20,21,22]. Therefore, the brain is one of the most active organs in the human body, and has a high metabolic rate. It is important to understand how the brain maintains tissue homeostasis and removes metabolic waste. The glymphatic system, first described in 2012, seems particularly important in this regard [23]. The term “glymphatic” refers to its function, which is similar to that of the lymphatic system in other organs, and the participation of glial cells in its structure [23]. However, reports from more than half a century ago seem to support the concept of the glymphatic system. It was then concluded that an important element of fluid circulation in the brain is lymphatic drainage due to the fact that CSF from the subarachnoid space drains into the cervical lymph system in the nasal cavity, in the orbita, and in the region of the jugular foramen. The glymphatic system, which has recently been described, is the missing component of the CSF flow route previously outlined [24,25]. The functioning of the glymphatic system is based on the bulk flow of CSF through the spaces that create it and cleanse the brain through this mechanism [26]. The glymphatic flow can be divided into three main stages [27]. First, CSF that is produced mainly in the choroid plexus flows from the subarachnoid space to the brain parenchyma [20]. This flow occurs via periarterial spaces called Virchow–Robin spaces [28]. Subsequently, AQP4 plays a crucial role in the second stage. AQP4 is a high-density water channel expressed on the end-feet of astrocytes surrounding vessels. This water channel mediates the flow from the periarterial spaces to the brain parenchyma, wherein the CSF from the interstitial space mixes with the interstitial fluid (ISF), and substances dissolve [26]. In the third stage, the ISF drains out of the brain through the perivenous spaces and, subsequently, into the peripheral lymphatic system (Figure 1) [26]. Efficient glymphatic flow and clearance depend on several factors. The following paragraph discusses AQP4 polarization, the sleep–wake state, and factors such as arterial pulsatility, respiration, and CSF pressure gradients [29]. They all contribute to unidirectional convective glymphatic flow, which plays a crucial role in maintaining brain homeostasis. Disturbances in the functioning of the glymphatic system have been described in the pathogenesis of AD and PD [30,31]. The examination of the glymphatic system in the context of neurodegeneration seems necessary to understand the pathogenesis of neurodegenerative disorders.

The glymphatic flow process can be categorized into three phases. The initial stages involve the inflow of CSF through the periarterial spaces. In the second stage, the CSF flows through AQP4 into the brain parenchyma, where CSF-ISF exchange occurs. With the flow of fluids through the brain parenchyma, soluble waste is washed out, and in the third stage, it drains out of the brain via perivenous CSF-ISF efflux and then drains into the peripheral lymphatic vessels.

## 4. Astrocytes and the Role of AQP4

Astrocytes outnumber neurons by over fivefold in the human brain [32]. The most important group of cells is inextricably related to CNS homeostasis, and plays a pivotal role in maintaining homeostasis. Astrocytes are characterized by a wide range of functions, from the regulation of brain energy metabolism through the clearance of neurotransmitters and ionic homeostasis to the modulation of synaptic activity and plasticity, which are only some of their functions [33]. Therefore, it is not surprising that functional impairment is observed in most groups of CNS diseases. Astrocytes are involved in the pathomechanisms of stroke, epilepsy, migraine, neuroinflammatory diseases, and neurodegenerative disorders [34]. The role of astrocytes in neurodegenerative disorders combines the glymphatic system and neuroinflammation. This is because of another fundamental function of astrocytes: they are essential for the effective functioning of the glymphatic system, and, as immunocompetent cells, play a crucial role in neuroinflammation, which is an integral part of neurodegeneration [26,35,36]. This section will discuss both functions of astrocytes, not only as coexisting, but also as potentially correlated.

As is well known, the functioning of the glymphatic system is based on the flow of CSF through the structures that form it. In this system, astrocytes are the gateway for CSF from the periarterial space to the brain’s interstitial space [26]. Thus, the disturbance of their function impairs glymphatic flow at the very beginning. It is impossible to discuss astrocytes’ function in this context without highlighting the role of the protein aquaporin 4 (AQP4). AQP4 is the most abundant water channel in the brain and is an integral part of the glymphatic system. The distribution of AQP4 is strongly polarized in the end-feet of astrocytes, which ensheath the blood vessels in the brain. This protein removes waste products from the brain, allowing glymphatic flow [37]. An animal model study showed that in mice with Aqp4 gene deletion, the clearance of the tracer from the brain interstitium was reduced by ~70% compared with that in wild-type animals [23]. Aβ is a protein associated with the pathogenesis of AD and is discussed in more detail in the next part of the review. As a result of the impairment of the glymphatic system due to the lack of AQP4, the clearance of Aβ from the brain decreases [38]. In animal models, the deletion of the Aqp4 gene has been shown to accelerate Aβ deposition [39]. Furthermore, a postmortem human brain tissue study in patients with AD revealed the loss of perivascular AQP4 localization, which was strongly associated with an increase in Aβ pathology [40]. Likewise, in this study of AD patients, it was observed that in the frontal cortical gray matter, the perivascular localization of AQP4 was significantly reduced compared to that in subjects without cognitive impairment. This was associated with an increase in Aβ levels, neurofibrillary pathological burden, and cognitive decline. To verify the significance of the postmortem findings, a mouse model lacking perivascular AQP4 localization was used. In this model, slowed CSF tracer influx and interstitial tracer efflux from the mouse brain as well as increased amyloid levels were observed [41]. In another study, an association was observed between the loss of the perivascular localization of AQP4 and the impairment of glymphatic flow, which resulted in increased Aβ levels in the mouse brain. What is particularly interesting is that this study compared the global Aqp4 gene deletion with the mislocalization of AQP4. In both cases, there was an increase in Aβ deposition, but the mislocalization of AQP4 had a more pronounced impact [42]. α-Syn, analogously to Aβ in AD, is a protein strongly associated with PD pathogenesis. A study in an animal model of PD showed that the impairment of the glymphatic system through the deletion of the Aqp4 gene reduced the clearance of injected α-Syn from the brain [5]. Interestingly, in mice overexpressing human A53T-α-syn, the expression and polarization of AQP4 were diminished and glymphatic activity was suppressed. Moreover, in these animals, AQP4 deficiency accelerated the accumulation of α-Syn and facilitated the loss of dopaminergic neurons, resulting in accelerated PD-like symptoms [5]. Thus, the role of AQP4 seems to significantly impact the formation of protein aggregates and the pathogenesis of neurodegeneration. However, the role of this protein in physiology and disease is not only to ensure efficient glymphatic flow and cleansing of the brain. The role of AQP4 in neurodegeneration is a potential link between glymphatic disorders and neuroinflammation (Table 1).

AQP4 polarization refers to its distribution, with a tenfold higher concentration in astrocytic end-foot membranes than in non-endfoot membranes [43,44]. This specific localization allows AQP4 to be in close contact with the perivascular space, thus facilitating CSF influx into the brain parenchyma [38]. Vascular polarized AQP4 expression is therefore crucial for the effective functioning of the glymphatic system [45]. The terms “astrogliosis”, “reactive astrocytes”, and “reactive astrogliosis” are used to describe the response of astrocyte cells in the form of biochemical, morphological, and metabolic changes to pathology, such as neurodegenerative diseases, trauma, ischemia, infection, or injury affecting the CNS [46]. Similar to neuroinflammation, astrogliosis is an essential element of the initial physiological response aimed at limiting damage and restoring homeostasis. However, through positive feedback loops driven by neuroinflammation, astrogliosis can impair astrocytic and neuronal functions and become a pathological and harmful process [47,48]. In response to pathological conditions, astrocytes participate in the production and release of an array of inflammatory mediators, such as chemokines, cytokines, complement factors, and reactive oxygen species (ROS), the participation of which is discussed in the context of the pathogenesis of neurodegenerative diseases [49]. Attention is drawn by the fact that in a mouse model of PD, a correlation was demonstrated between the increasing accumulation of α-Syn aggregates and the expansion of reactive astrogliosis [50]. Reactive astrocytes can participate in inflammatory responses and may be involved in both neuroprotective and neurodegenerative functions in neurodegenerative diseases. However, the knowledge in this area is still limited [51]. What is particularly important from the point of view of the discussed problem is the abnormal expression and polarization of AQP-4 in the conditions of astrogliosis and the potential consequences associated with it in the form of impaired glymphatic clearance. In reactive astrocytes, abnormalities in AQP4 polarization are observed, manifesting as an increase in AQP4 density in parenchymal membranes or a decrease in density in end-foot membranes [52]. In patients with idiopathic normal-pressure hydrocephalus, there was a significant correlation between the degree of astrogliosis and the reduction in AQP4 polarization at the astrocytic end-feet [53]. A study on an animal model of AD showed that animals develop astrogliosis and the localization pattern of AQP4 changes [54]. In a mouse PD model, astrogliosis and AQP4 depolarization contributed to glymphatic dysfunction [55]. Disturbances in the expression and localization of AQP4 may be associated with impaired glymphatic flow and may contribute to the pathological processes of neurodegeneration [56]. This issue requires further research into the role of astrogliosis in neurodegeneration, with particular emphasis on its impact on AQP4 polarization and glymphatic flow. However, the discussed AQP4 polarization is not the only factor associated with neuroinflammation.

The potential impact of AQP4 on neuroinflammatory processes is under research, but there are conflicting points of view. It has been shown that AQP4 may play a pro-inflammatory role in the PD model. Wild-type (WT) and Aqp4 deletion mice (Aqp4-/- mice) were injected with 1-methyl-4-phenylpyridinium (MPP+, a parkinsonogenic toxin). Interestingly, in the WT mice, the expression levels of microglial-activating genes increased, which was not observed in the Aqp4-/- mice. Moreover, in the WT mice, MPP+ injections caused the upregulation of AQP4 and the swelling of astrocytic end-feet [57]. On this basis, it was concluded that AQP4 plays a pro-inflammatory role via the activation of microglia and the swelling of astrocytes [57]. The involvement of AQP4 in the inflammatory response of astrocytes has also been investigated. An increase in the levels of proinflammatory cytokines (TNF-α and IL-6) after the addition of LPS was measured in astrocyte cultures from WT and AQP4-null mice. Significantly greater cytokine release and neuroinflammation were observed in the WT cultures than in the AQP4-null samples [58]. However, it seems impossible to define AQP4 as a definitive pathogenic factor for neurodegeneration. There is no doubt about its crucial role in the efficient functioning of the glymphatic system, which is essential for maintaining brain homeostasis [26,38]. Moreover, studies in an animal model of AD have shown that in APP/PS1 transgenic mice, AQP4 deletion exacerbates cognitive deficits and is associated with an increase in Aβ accumulation [39]. This is important in the context of neuroinflammation, because Aβ plays an essential role in promoting neuroinflammation via microglial immune activation and the induction of neurotoxic astrocytes [59]. Therefore, AQP4 may be a therapeutic target for neurodegenerative diseases that may influence neuroinflammation in their course. However, due to the complexity of the problem and opposing positions, whether therapy should strive to increase or decrease its expression seems complicated at this point and requires further research. The influence of AQP4 on neuroinflammatory and neurodegenerative processes seems to depend on the type of disease and its stage [37]. Moreover, in addition to its expression, its polarization is also essential. Defining AQP4 as unequivocally pro-inflammatory or anti-inflammatory is not possible; however, expanding the research on its role in the specific stages of various diseases may be valuable (Figure 2).

It appears that AQP4 is a significant common component in the processes of neuroinflammation and glymphatic clearance. In this context, a particularly important issue is reactive astrogliosis, which is linked to neuroinflammation and could potentially affect the efficiency of glymphatic clearance through the negative impact of AQP4 polarization. The expansion of reactive astrogliosis may be influenced by the accumulation of αSyn, which is intrinsically linked to the process of neurodegeneration and is also subject to glymphatic clearance. The role of AQP4 in neuroinflammation remains a matter of debate. The pro-inflammatory and anti-inflammatory nature of this protein has been subject to conflicting opinions. Nonetheless, there is no ambiguity regarding its significant impact on inflammatory processes and its pivotal role in the glymphatic system, rendering AQP4 a promising target for further investigation and a potential avenue for therapeutic interventions.

**Table 1 cells-13-00286-t001:** Multiple roles of AQP4 in neurodegeneration including a potential connection between glymphatic disorders and neuroinflammation.

Multiple Roles of AQP4 in Neurodegeneration	Description	References
Loss of the perivascular localization of AQP4	Results in glymphatic impairment and in an increase in Aβ level	Pedersen, T.J. et al., 2023 [42]
Deletion of the Aqp4 gene	Results in reduced clearance of injected α-Syn from the brain	Zhang, Y. et al., 2023 [5]
Significant reduction of perivascular localization of AQP4	Observed in postmortem studies of patients with AD compared to subjects without cognitive impairment	Simon, M. et al., 2022 [41]
AQP4 deficiency	In an animal model accelerated the accumulation of α-Syn, facilitated the loss of dopaminergic neurons and, accelerated PD-like symptoms	Zhang, Y. et al., 2023 [5]
Pro-inflammatory role of AQP4	Activation of microglia and swelling of astrocytes were observed in an animal model of PD	Prydz, A. et al., 2020 [57]
Involvement of AQP4 in the inflammatory response of astrocytes	Increase in the level of proinflammatory cytokines measured in astrocyte cultures was observed	Li, L. et al., 2011 [58]
AQP4 deletion	Results in exacerbation of cognitive deficits and was associated with an increase in Aβ accumulation in an animal model of AD	Xu, Z. et al., 2015 [39]

## 5. Protein Aggregates as a Driver of the Vicious Circle of Impaired Glymphatic Clearance and Neurodegeneration

AD and PD belong to a group of diseases known as proteinopathies. Pathological protein aggregation and the protein aggregates formed during this process are hallmarks of neurodegeneration. As a result, proteins that originally play an essential and physiological role in the cell may gain toxicity and lose the function that drives neurodegeneration [11,60]. In PD, the protein that forms aggregates, termed Lewy bodies, is α-Syn [12]. The pathological deposition of Aβ has been observed in AD [13]. The participation of these proteins in neurodegenerative processes has been emphasized in the literature [12,13]. However, the potential correlation between protein aggregates in the context of neuroinflammation and glymphatic clearance remains unclear. Microgliosis, which is a component of neuroinflammation, is an inherent element of neurodegeneration. Both aggregated α-Syn and Aβ can induce microglial activation and neuroinflammation [60]. It has been shown that microglia can bind Aβ via the CD36 scavenger receptor. The binding of Aβ to the CD36 receptor plays an important role in proinflammatory events, leading to microglial activation and the production of pro-inflammatory cytokines [61]. Moreover, pro-inflammatory cytokines influence Amyloid Precursor Protein (APP) levels. Neuroinflammatory cytokines elevate the expression level of APP and, consequently, the production of Aβ increases [62]. Therefore, Aβ production and microglial activation appear to result in a positive feedback loop, leading to ongoing inflammation [63]. Similarly, in PD, α-Syn induces microglial activation via Toll-like receptors (TLRs) and the release of proinflammatory cytokines, which aggravate CNS inflammation [64]. α-Syn also participates in activating inflammasomes, which are macromolecular protein complexes playing an important role in the functioning of the innate immune system. In PD, misfolded α-Syn triggers persistent NLRP3 inflammasome activation and, through the release of proinflammatory cytokines, results in neuronal cell death [65]. In this case, the processes are also closed in a positive feedback loop, driving neurodegeneration. Studies have shown that microglial activation promotes the phosphorylation and aggregation of α-Syn and upregulates neuronal TLR2 and TLR4 [66]. Moreover, activated microglia play a significant role in prion-like α-Syn propagation between cells [67]. Therefore, there is considerable evidence that protein aggregates play a crucial role in neurodegeneration in the context of neuroinflammation. However, this is not the only area that influences the pathological cascade. When discussing protein aggregates, it is impossible to ignore the pathway by which the brain cleanses itself. The first description of the glymphatic system in 2012 emphasized its role in Aβ clearance. An animal model has shown that glymphatic flow contributes to the clearance of soluble Aβ from the brain; the mice were injected with 125I-amyloid β1–40. In animals with impaired functioning of the glymphatic system due to the global knockout of the Aqp4 gene, clearance was reduced by ~55% compared to the wild-type controls [23]. The role of the glymphatic system in protein clearance has been confirmed in several studies. It was shown that owing to the weakening of glymphatic clearance efficiency, which was correlated with advancing age, the clearance of intraparenchymally injected Aβ was impaired by 40% [68]. This is consistent with the incidence of neurodegenerative diseases, including AD, which increases with age and most often affects the elderly [1]. Due to the limitations of human research, most studies to directly demonstrate the activity of the glymphatic system have been conducted on animal models. However, a non-invasive magnetic resonance imaging technique allows us to check the activity of the glymphatic system in vivo in humans. This method, called diffusion tensor imaging (DTI), is based on measuring the diffusive movements of water molecules in the extracellular space of imaged tissues [69]. The study included 21 patients with AD and 36 healthy subjects who underwent diffusion tensor imaging (DTI) and positron emission tomography (PET) using the Aβ tracer 11C-PiB and tau/inflammatory tracer 18F-THK5351. It was shown that the impaired functioning of the glymphatic system demonstrated by DTI imaging is correlated with the increased deposition of Aβ and tau, and inflammation of the nervous system [70]. Similar conclusions have been drawn from previous studies on the influence of glymphatic flow on the clearance of α-Syn. In a mouse model of PD, decreased AQP4 expression accelerated the pathological deposition of α-Syn and aggravated the loss of dopamine neurons. On this basis, it was concluded that the probable impairment of glymphatic system function is involved in the progression of PD through increased α-Syn deposition [71]. Importantly, the relationship between glymphatic clearance and α-Syn accumulation is bidirectional. The study, similar to the one mentioned earlier, confirmed that AQP4 deficiency and the resulting glymphatic insufficiency accelerated the accumulation of α-Syn and facilitated the loss of dopaminergic neurons [5]. However, an equally important second conclusion is that the overexpression of A53T- α-Syn reduced the expression/polarization of AQP4 and suppressed glymphatic activity [5]. Thus, weakened glymphatic clearance contributes to the accumulation of α-Syn. In addition, the accumulation of α-Syn contributes to the impairment of glymphatic system function, which, once again, closes the processes in a vicious circle. Therefore, it can be concluded that protein aggregates act as a positive feedback loop in both neuroinflammatory processes and the impairment of glymphatic system function. These self-perpetuating processes may converge and promote neurodegeneration. Therefore, consideration should be paid to whether new potential therapeutic activities should focus on all of the above-mentioned aspects, the convergent point of which is pathological protein aggregation.

## 6. Peripheral Inflammatory Cells in the Brain

When considering neuroinflammation, the cells responsible for the systemic immune response must be considered. The main population of immune cells in the central nervous system is microglia, which are macrophages that populate this area during fetal life [72,73]. However, many pathological conditions, such as multiple sclerosis, stroke, or traumatic brain injury, can cause an influx of systemic immune cells, such as monocytes, neutrophils, macrophages, and T lymphocytes, into the CNS area [74,75,76]. The infiltration of the brain by immune cells is also explicitly associated with neurodegenerative diseases such as AD and PD [77,78]. The influence of T lymphocytes appears to be significant in the management of PD. Cytotoxic and helper T lymphocytes, but not B cells, have been observed in postmortem studies near dopaminergic areas [79]. This was also confirmed in a mouse model study in which the selective nigrostriatal infiltration of CD4+ and CD8+ lymphocytes was observed in MPTP-treated mice [79]. In addition, PD patients showed a significant increase in the proportion of activated lymphocytes with a concomitant decrease in naive lymphocytes, indicating their peripheral activation in PD [80]. Moreover, it has been proven that α-Syn can be an antigen for T lymphocytes, thereby activating the immune response [81]. Autoantibodies to α-Syn are believed to be involved in the clearance pathway, among other processes, indicating a significant link to the glymphatic system and a protective role in the progression of PD [82]. However, it is not only cell-specific responses that are significant in neuroinflammation associated with neurodegeneration. It has been shown that neutrophils, the most abundant group of leukocytes, have a significant association with the progression of AD [83,84]. In a mouse model of AD, it has been observed that Aβ plaques act on neutrophils chemotactically by stimulating their migration across the BBB [85]. Neutrophils had a significantly higher motility rate and migration velocity in AD mice than WT mice [83,85]. Human studies have also confirmed the involvement of neutrophils in AD progression. In AD patients, neutrophils have been found in the areas of the cerebellum, temporal cortex, and hippocampus in both the parenchyma and vascularities of these areas [84,86,87]. Neutrophils secrete characteristic proteins during the immune response, such as CAP37, neutrophil elastase, and cathepsin G [88,89,90]. The presence of cathepsin G and elastase has also been confirmed in microglia, one of the primary components of neuroinflammation [91,92]. Moreover, CAP37 has been discovered to be a molecule that activates microglia cells and causes their polarization to the M1 pro-inflammatory type [93,94]. This information indicates the interaction between microglia and neutrophils in the progression of neuroinflammation and the possibility of microglial polarization to the pro-inflammatory type by incoming neutrophils. Neutrophils are characterized by the formation of neutrophil extracellular traps (NETs) composed of extracellular strings of DNA linked to histones and several neutrophil granule proteins [95]. NETs are known for their strong pro-inflammatory effects during the peripheral immune response [96]. There was no difference in the neutrophil influx into the CNS during AD. Studies using immunostaining and immunoassays have shown that NETs occur in large and small cerebral vessels during AD. The same study also confirmed that the accumulation of myeloperoxidase (MPO), an enzyme that generates ROS, occurs near Aβ plaques and is stimulated by an increase in the number of intravascular neutrophils [86]. The activation and influx of immune cells into the brain are a double-edged sword. Many processes, such as autoantibodies against α-Syn, have protective effects. However, in the long term, the influx and activation of immune cells stimulate neuroinflammation. What was pointed out above could result in the dysfunction of the glymphatic system and, thus, the progression of neurodegenerative diseases.

## 7. Gut–Brain Axis

Microbiota are trillions of microorganisms that inhabit the human body, especially the gastrointestinal tract [97]. It was concluded that these organisms are among the most important regulators of gut–brain axis function. The term microbiota–gut–brain axis was created, and disturbances in its functioning have been shown to play a potential role in many disorders, such as schizophrenia, anxiety, and neurodegenerative diseases, including AD and PD [98]. There are many pathways for bidirectional communication that are mediated by microorganisms. Communication pathways can be distinguished as immune, including immune cells and the cytokines secreted by them; neural, made by the vagus nerve and enteric nervous system; and communication paths relying on neurotransmitters, intestinal peptides, and bacterial metabolites [99]. Dysbiosis refers to compositional and functional alterations in the microbiome, increasing proinflammatory species, or the loss of beneficial organisms [100,101]. Interestingly, dysbiosis has been associated with neuroinflammation occurring in both AD and PD [102]. The cascade of events in which the microbiome plays a significant role includes a pro-inflammatory state in the gut, which increases its permeability, and a systemic inflammatory response, which, in turn, affects neuroinflammation [102]. Lipopolysaccharide (LPS) is the main component of the outer membrane of Gram-negative bacteria. This substance can trigger systemic inflammation, resulting in the release of pro-inflammatory cytokines [103]. The systemic administration of LPS to transgenic mice induces neuroinflammation and results in more significant amyloid deposition and tau pathology [104]. Another study indicated that peripheral LPS injection affected the NLRP3 inflammasome, increasing pro-inflammatory cytokine expression and microglial activation [105]. It is worth focusing on the role of NLRP3, which plays a role in neuroinflammation in CNS diseases, as in the previously mentioned PD [65]. NLRP3 is involved in the regulation of the gut–brain axis. Through NLRP3 signaling, intestinal bacteria modulate inflammatory pathways that affect brain homeostasis [106]. Moreover, a postmortem study showed that in brain tissue from patients with AD, LPS and Gram-negative *E. coli* fragments colocalize with amyloid plaques [107]. Dysbiosis resulting in an increased number of Gram-negative bacteria, including Bacteroides, may contribute to AD pathology by affecting neuroinflammation and pathological protein aggregation [103]. Similarly, the gut microbiota has been shown to influence neuroinflammation and α-Syn aggregation in a mouse model of PD [108,109]. Moreover, after colonizing α-Syn-overexpressing mice with microbiota from PD patients, motor function impairment increased compared to microbiota transplants from healthy human donors [108].

The importance of microbiota in the development of CNS pathology seems to be significant. Their multi-aspect impact on neurodegeneration and neuroinflammatory processes is beyond the scope of this study and has been described in many publications [102,110,111,112]. However, microbiota may be another point of convergence for neuroinflammation and glymphatic dysfunction. Dysbiosis impacts the severity of neuroinflammation, and protein aggregation ultimately affects the glymphatic system. However, at least one additional interesting issue is worth discussing. The blood–brain barrier (BBB) interface consists of blood vessels and interstitial fluid throughout the brain. It consists of many types of cells, whose common task is to protect and regulate the brain’s microenvironment [113]. The BBB and glymphatic system cooperate to clear interstitial solutes such as Aβ from the brain [114]. It is essential that they influence each other and that BBB disruption affects the glymphatic system, among others, through neuroinflammation. Inflammation decreases convective flow and consequently negatively affects the rate of CSF-to-ISF turnover, resulting in impaired glymphatic clearance [114,115]. This is important regarding microbiota, not only because of the inflammation they may be a source of, but also because of the potential impact on the BBB [116]. Short-chain fatty acids (SCFAs) are gut microbiota products produced mainly by fermentation [117]. These compounds may also affect the integrity of the BBB. Butyrate, which belongs to the SCFA group, upregulates tight junction proteins and reduces BBB permeability [118]. It has also been shown that butyrate attenuates pro-inflammatory cytokine expression in microglia and has a beneficial effect on neuroinflammation in aging mice [119]. Another study indicated that germ-free mice showed reduced expression of the tight junction proteins, occludin and claudin-5, which are involved in maintaining proper BBB function. The exposure of animals to gut microbiota increased the expression of tight junction proteins and decreased BBB permeability [116]. Therefore, there seems to be a possible connection between microbiota and the functioning of the glymphatic system [20]. Further research is required to determine the direct and indirect effects of microbiota on glymphatic system function. However, it can be concluded that microbiota, and specifically dysbiosis, play a role in neurodegeneration and accompanying neuroinflammation. The intricate significance of dysbiosis, BBB permeability, and other crucial factors, including the glymphatic system and neuroinflammation, as discussed in the main text, and their interconnections, are elucidated in Figure 3.

The figure presents the issues discussed in the main text that are important in the process of neurodegeneration. At the top of the frames, the main factors discussed are outlined in distinct colors. These colors are assigned to specific elements such as neuroinflammation (red), protein aggregates (violet), glymphatic system impairment (dark blue), NLRP3 signaling (cyan), dysbiosis (green), and BBB permeability (yellow). The colors mentioned are subsequently used in the text in the bullets of other frames to illustrate the interdependence/impact of the components.

## 8. Conclusions

In conclusion, neuroinflammation and impaired glymphatic system function contribute to the progression of neurodegenerative disorders and appear to be important factors in their etiology. Protein aggregates are characteristic features of neurodegeneration, and are correlated with glymphatic clearance and neuroinflammation. Astrocytes also play a crucial role in inflammatory responses in the central nervous system and are essential for the effective functioning of the glymphatic system. The gut–brain axis and microbiome may impact the neuroinflammatory and neurodegenerative processes.

## Figures and Tables

**Figure 1 cells-13-00286-f001:**
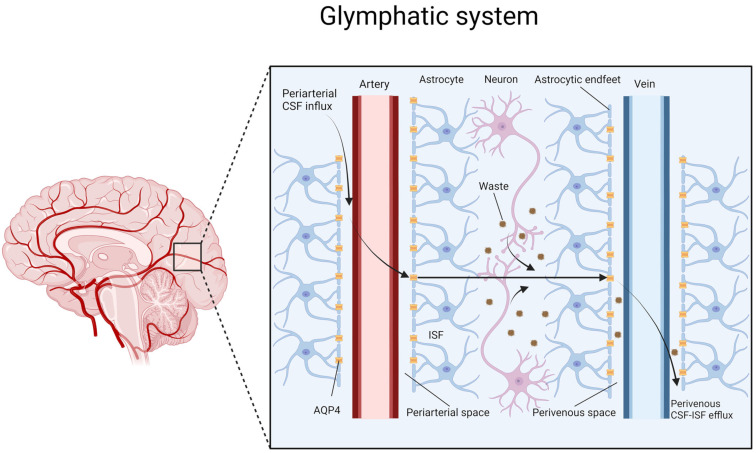
Schematic representation of the glymphatic system.

**Figure 2 cells-13-00286-f002:**
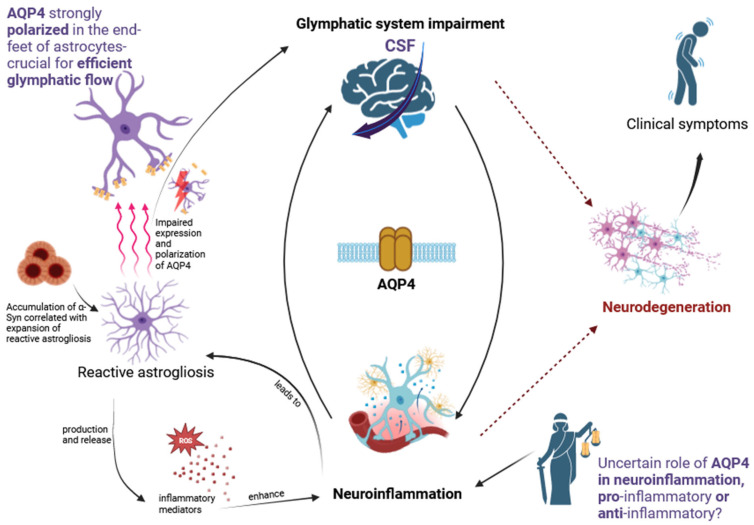
Role of AQP4 linking the processes of neuroinflammation and glymphatic impairment in neurodegeneration.

**Figure 3 cells-13-00286-f003:**
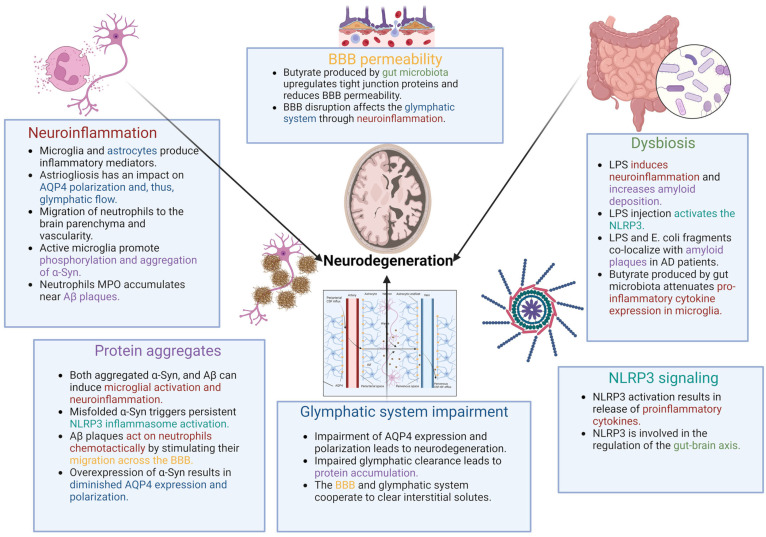
Influence of individual factors and their interrelationships in the pathogenesis of neurodegeneration.

## Data Availability

No new data were created or analyzed in this study. Data sharing is not applicable to this article.

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
