# Peer review of "Glymphatic System Pathology and Neuroinflammation as Two Risk Factors of Neurodegeneration"

_cells, 2024, doi:10.3390/cells13030286_

Round 1
Reviewer 1 Report
Comments and Suggestions for Authors
The glymphatic system is an emerging system for waste clearance in the Central Nervous System that may have important clinical significance in the onset and development of neurodegenerative disease due to the accumulation of toxic aggregates.
The authors propose a clear overview of the impact of glymphatic stystem on neurodegenerative diseases concluding that protein aggregates correlate with glymphatic clearance and neuroinflammation reporting the latest researches in the field, thus providing an update of the state of the art.
In my opinion, the aim of the review is clear and the reference are appropriates, so I have no suggestions for author.
Author Response
REVIEWER 1:
The glymphatic system is an emerging system for waste clearance in the Central Nervous System that may have important clinical significance in the onset and development of neurodegenerative disease due to the accumulation of toxic aggregates.
The authors propose a clear overview of the impact of glymphatic stystem on neurodegenerative diseases concluding that protein aggregates correlate with glymphatic clearance and neuroinflammation reporting the latest researches in the field, thus providing an update of the state of the art.
In my opinion, the aim of the review is clear and the reference are appropriates, so I have no suggestions for author.
ANSWER: Thank you for your time to review our manuscript.

Reviewer 2 Report
Comments and Suggestions for Authors
This is a solid review but rather heavy to read. I suggest to add a couple of figures. One good and informative would be a schematic presentation of the glymphatic anatomy.
The authors would demonstrate deeper visions, if they added more historical aspects, see the attached PDF file from Nature 2015: 524, 415 (Mezey & Palkovits). This concerns specifically page 3, lines 81-82 and page 7, lines 258-259.
One interesting aspect to consider regarding microbiota behind the brain problems is that why both the wife and the husband been married for tens of years and share the same microbiota, do very seldom get AD or PD - usually only one of those (if either).
Minor
Page 8, line 312: an epitope should be an antigen
Page 8, line 340: vascular neutrophils. What type of neutrophils are they - inside the vasculature marginated or in the tissue close to vasculature?

Please use an uniform presentation throughout the text (for example alfa synuclein has now been written in 3 different ways) and if you are using AD and PD, use them throughout after their introduction.
Author Response
REVIEWERS COMMENTS:
REWIEVER 2:
This is a solid review but rather heavy to read. I suggest to add a couple of figures. One good and informative would be a schematic presentation of the glymphatic anatomy.
ANSWER 1: Thank you for your time to review our manuscript. Due to your recommendation, we added more figures to the article, one of them (new Fig. 1) is a schematic presentation of the glymphatic anatomy.
The authors would demonstrate deeper visions, if they added more historical aspects, see the attached PDF file from Nature 2015: 524, 415 (Mezey & Palkovits). This concerns specifically page 3, lines 81-82 and page 7, lines 258-259.
ANSWER 2: Thank your for your comment. We have changed the manuscript in relation to your comment, also adding the aspects described in the article you mentioned. Thank you.
One interesting aspect to consider regarding microbiota behind the brain problems is that why both the wife and the husband been married for tens of years and share the same microbiota, do very seldom get AD or PD - usually only one of those (if either).
ANSWER 3: Thank you for your comment. We have searched the literature and have not found any clear evidence to adress your question recently describing the exact mechanism of your proposed problem. Due to the fact that microbiota is not the main point of our review, we would like to pay more attention to the relation between glymphatic system and microbiota. Your commentary is very intriguing and we would like to explain it more in another review – maybe together?
Minor
Page 8, line 312: an epitope should be an antigen
ANSWER: Thank you, it has been corrected.
Page 8, line 340: vascular neutrophils. What type of neutrophils are they - inside the vasculature marginated or in the tissue close to vasculature?
ANSWER: Thank you for your comment. We have corrected the name of the neutrophils on “intravascular neutrophils”
Please use an uniform presentation throughout the text (for example alfa synuclein has now been written in 3 different ways) and if you are using AD and PD, use them throughout after their introduction.
ANWER: Thank you, of course, you are wright. We have checked the whole manuscript and unified the whole text.

Reviewer 3 Report
Comments and Suggestions for Authors
In their work, the authors provided a literature review focused on identifying potential correlations between the pathogenesis of neurodegenerative diseases and the functioning of the glymphatic system. The manuscript contains crucial data on the role of the glymphatic system and signaling components of neuroinflammation in the mechanism of neurodegenerative disease pathogenesis. The authors have compiled an interesting set of information, and its purpose and conclusions are presented in a clear manner. The manuscript is well-written, and the literary sources are thoroughly examined.
However, there are a few comments:
- Figure 1 lacks a caption.
- Figure 1 needs to be detailed and colored. It would be particularly illustrative to include some of the molecular and cellular components mentioned in the manuscript related to the pathogenesis of neurodegeneration (aberrant proteins, AQP4, the inflammation cascade, etc.).
- In lines 34-35, there is an incomplete discussion on the pathogenesis of neurodegenerative diseases, lacking information on aberrant proteins.
- The manuscript's abstract lacks information on AQP4, despite its significant focus in the paper.
- It would be beneficial if the authors could prepare 2-3 schematic figures to facilitate the understanding of the information presented in sections 4, 5, 6, and 7.
- The sentence in lines 411-412 appears redundant in the conclusion.
Author Response
REVIEWERS COMMENTS:
REVIEWER 3:
In their work, the authors provided a literature review focused on identifying potential correlations between the pathogenesis of neurodegenerative diseases and the functioning of the glymphatic system. The manuscript contains crucial data on the role of the glymphatic system and signaling components of neuroinflammation in the mechanism of neurodegenerative disease pathogenesis. The authors have compiled an interesting set of information, and its purpose and conclusions are presented in a clear manner. The manuscript is well-written, and the literary sources are thoroughly examined.
ANSWER: Thank you for your time to review our manuscript and these comments.
However, there are a few comments:
- Figure 1 lacks a caption.
ANSWER: Thank you for your comment. We have rebuilt Fig.1 due to your comments.
- Figure 1 needs to be detailed and colored. It would be particularly illustrative to include some of the molecular and cellular components mentioned in the manuscript related to the pathogenesis of neurodegeneration (aberrant proteins, AQP4, the inflammation cascade, etc.).
ANSWER: Thank you for this comment. Fig 1. has been rebuilt due to your comments. We also created Fig. 2 to describe in a colourful way the suggestions, you mentioned.
- In lines 34-35, there is an incomplete discussion on the pathogenesis of neurodegenerative diseases, lacking information on aberrant proteins.
ANSWER: Thank you, we have corrected this part of the manuscript.
- The manuscript's abstract lacks information on AQP4, despite its significant focus in the paper.
ANSWER: Thank you for your comment. We have rebuilt the abstract adding the information about AQP4 as much as we could due to abstract size limitations.
- It would be beneficial if the authors could prepare 2-3 schematic figures to facilitate the understanding of the information presented in sections 4, 5, 6, and 7.
ANSWER: Thank you. Due to your comments, we have built also Fig 3 summarizing parts of the manuscript, you mentioned.
- The sentence in lines 411-412 appears redundant in the conclusion.
ANSWER: Thank you. We have changed it due to your suggestion.

Round 2
Reviewer 2 Report
Comments and Suggestions for Authors
The manuscript is now suitable for publication.
Reviewer 3 Report
Comments and Suggestions for Authors
The authors have compiled an interesting set of information, and its purpose and conclusions are presented in a clear manner